# Effect of a Composite Alginate/Grape Pomace Extract Packaging Material for Improving Meat Storage

**DOI:** 10.3390/ijms242115958

**Published:** 2023-11-03

**Authors:** Antonella Maria Aresta, Nicoletta De Vietro, Jennifer Gubitosa, Vito Rizzi, Ilaria De Pasquale, Paola Fini, Pinalysa Cosma, Maria Lucia Curri, Carlo Zambonin

**Affiliations:** 1Dipartimento di Bioscienze, Biotecnologie e Ambiente, Università degli Studi “Aldo Moro” di Bari, Via Orabona, 70126 Bari, Italy; antonellamaria.aresta@uniba.it (A.M.A.); carlo.zambonin@uniba.it (C.Z.); 2Dipartimento di Chimica, Università degli Studi “Aldo Moro” di Bari, Via Orabona, 70126 Bari, Italy; vito.rizzi@uniba.it (V.R.); pinalysa.cosma@uniba.it (P.C.); maria.curri@uniba.it (M.L.C.); 3Consiglio Nazionale delle Ricerche CNR-IPCF, UOS Bari, Via Orabona, 70126 Bari, Italy; i.depasquale@ba.ipcf.cnr.it (I.D.P.); p.fini@ba.ipcf.cnr.it (P.F.)

**Keywords:** food packaging materials, alginic acid, grape pomace polyphenolic extract, red meat, biogenic amines, solid-phase microextraction/gas chromatography–mass spectrometry, waste management

## Abstract

The development of food packaging materials that reduce the production of plastic, preserving at the same time the quality of food, is a topic of great interest today for the scientific community. Therefore, this article aims to report the effectiveness of an eco-friendly packaging material based on alginic acid and grape pomace extract from *Vitis vinifera* L. (winemaking by-products) for storing red meat in a domestic refrigerator. Specifically, biogenic amines are considered “sentinels” of the putrefactive processes, and their presence was thus monitored. For this purpose, an experimental analytical protocol based on the use of solid-phase microextraction coupled with gas chromatography–mass spectrometry was developed during this work for the determination of six biogenic amines (butylamine, cadaverine, isobutylamine, isopentylamine, putrescine, and tyramine). Moreover, by combining the analytical results with those of pH and weight loss measurements, differential scanning calorimetry, and microbiological analysis, it was proved that the studied materials could be proposed as an alternative packaging material for storing foods of animal origin, thus lowering the environmental impact according to sustainability principles.

## 1. Introduction

Amines are low molecular weight organic bases produced in animals, plants, or microorganisms after specific metabolic pathways [1,2]. In foods, they can be naturally occurring or produced by microbial enzymatic amino acids decarboxylation, so it is possible to detect variable quantities of biogenic amines (BAs) that could pose a potential risk to consumer safety. The total amount of the formed BAs essentially depends on the nature of the food and the microorganisms [3]. Unfortunately, the complexity of the reactions in which the BAs are involved renders it difficult to identify a toxicity threshold, which ultimately depends on the efficiency of the detoxification system changes related to individual and physiological variables. Among the most important foodborne syndromes caused by the ingestion of foods containing these compounds are histamine intoxication, known as “Scombroid syndrome”, due to its presence in fish and meat, and tyramine intoxication, so-called “Cheese syndrome”, recognized in several epidemiological studies [4,5,6,7]. However, even when their blood concentration does not reach levels considered toxic, the presence of BAs can constitute a risk for the possible formation of nitrosamines, powerful oncogene agents that are formed by the reaction between nitrous acid and secondary amines in the gastric acidic environment [8]. For the potential health risk that BAs should pose to the consumers, they represent important indicators of food quality having animal origin. Indeed, their presence can be related to the hygienic quality of the raw materials and a state of freshness; in other cases, their concentrations in foods indicate the cause–effect relationship relating to improper production and/or storage conditions. For example, the most frequent amines found in the meat are spermidine, spermine, putrescine, cadaverine, and tyramine. Cadaverine, putrescine, tyramine, and isopentylamine are generally evaluated to estimate the state of conservation of meat [9,10]. Butylamine and isobutylamine, deriving from butyric acid, are responsible for the characteristic smell of “rotten fish” [11], and, therefore, their increase is, once again, the spoilage index of food of animal origin.

Packaging conditions have an influence on the production of these compounds during storage, and proper packaging materials can provide significant benefits to both producers and consumers by representing an essential method for improving food product safety at each supply chain level. It is worth mentioning that to face the problem of plastics, among different packaging materials, natural polysaccharides such as chitosan and sodium alginate (SA) have been widely proposed in this field due to their high barrier to gases, antioxidant, and antimicrobial properties [12].

In our recent work [13], to confer to an alginate-based packaging additional prop-erties, an aqueous extract from agrifood wastes, e.g., grape pomace (*Vitis vinifera* L.) was used to form hybrid films SA-based rich in polyphenols, well-known antioxidants, and sunscreen active compounds, lignin, and proteins [14], cost-effective, and antimicrobial [15].

A modified protocol was followed for preparing water-resistant SA-based films by adopting both external and internal gelation methods. More specifically, as a first step, the SA hydrogel was mixed with a 2.5% (*w*/*v*) CaCl_2_ solution and then placed in an oven to remove water; thus, the solid-state film formation occurred. The obtained free-standing SA-based films were placed in contact with a 5% (*w*/*v*) CaCl_2_ solution.

Depending on the applications, both methods would offer SA-based films constituted by polymer chains differently packed and water-resistant if compared with those realized without using CaCl_2_ [13]. In this particular case study, two phenomena occurred simultaneously: (i) hydration and (ii) the further crosslinking reaction with Ca^2+^, favored by the high CaCl_2_ concentration [13]. 

Strong coulombian interactions occurred between Ca^2+^ ions and the alginate carboxylic groups, compacting the polymeric network and making it less hydrophilic and less permeable to water [13]. 

The film was stable at high temperatures and not pH-responsive. Only highly concentrated salt-based solutions negatively affected the proposed packaging, inducing a large swelling [13].

In this work, the application of these films based on sodium alginate enriched with grape pomace polyphenolic extract (SA/GPPE) was evaluated for storing red meat (e.g., beef), proposing the development of an eco-friendly material that would respect sustainability and bio-circular principles, valorizing the alternative reuse of wastes. Accordingly, the grape pomace by-product management, whose presence in the environment could induce harmful effects [16], represents an important topic to be considered. For these reasons, its reuse for food industry applications represents a way to reduce the environmental impact [13,14], valorizing, at the same time, a waste to be recovered. Moreover, the presence of polyphenols in packaging could preserve the quality of food, acting also as antimicrobial agents [17]. Pseudomonas, Enterobacteriaceae, enterococci, and lactobacilli are devoted to the BAs production [9,10].

To demonstrate the effectiveness of the proposed material, the biogenic amine presence and the antimicrobial activity were evaluated during the red meat storage at different contact times in a domestic refrigerator. Chromatographic techniques represent the preferred approach for the determination of BAs [18] after derivatization. Therefore, sample preparation can be very time-consuming and requires the use of organic solvents. Solid-Phase MicroExtraction (SPME), coupled with gas chromatography–mass spectrometry (GC–MS), is an innovative approach that does not involve toxic organic solvents and offers additional sensitivity for the determination of BAs. This procedure uses the direct immersion (DI) of the fiber into a suitable solution containing the derivatizing agent, that is, isobutyl chloroformate (IBCF).

So, the DI-SPME procedure, coupled with an optimized GC–MS protocol, has been successfully applied for the quantitative determination of six BAs, namely, butylamine, cadaverine, isobutylamine, isopentylamine, putrescine, and tyramine in beef wrapped with the developed SA/GPPE-based packaging material. For comparison, the same amount of meat was preserved with a commercial polyethylene (PE) film, commonly used to preserve food, and with lone SA film.

All samples were kept in a domestic refrigerator at 8 °C and, at regular intervals of time (0, 2, and 6 days), for each of them, the quantitative evolution of the selected BAs pattern produced from meat during its physiological aging process was monitored.

The pH and weight loss measurements, differential scanning calorimetry (DSC), and microbiological analysis were also performed on beef samples wrapped in the three packaging materials at each storage time at the established temperature.

## 2. Results and Discussions

### 2.1. SPME/GC–MS Analysis for BAs Detection

In the combined SPME and derivatization procedure, numerous parameters require optimization, such as the choice of fiber absorbent material, ionic strength, pH of the solution, extraction time, and derivation conditions. In this study, previously optimized parameters were used with positive results [19].

First, a mix standard (75 µg/mL) in a vial was subjected to the SPME and derivatization procedure, as described in the Section 3. Then, the PA-SPME fiber was carefully removed from the vial and directly inserted into the GC injector for the GC–MS analysis. Retention times and spectra (Figure 1) of the BAs IBCF derivatives, not present in the NIST library, were acquired. Analyzing the acquired spectra, the m/z ions shown in Table 1 (for more details, see Materials and Methods, Section 3.2.2) were selected to obtain the extracted ionic chromatograms (XICs) used to derive the method validation parameters shown in Table 2.

Figure 2 shows the XIC of a standard BAs mix solution (concentration level: butylamine (RT = 6.68 min), isobutylamine (6.06), and isopenthylamine (7.48) 0.05 µg/mL; cadaverine (12.95), putrescine (12.52), and tyramine (14.18) 0.5 µg/mL) (Figure 2A), and of the as purchased (t_0_) beef sample (Figure 2B). 1,7-diaminoheptane (IS) (14.04) is always in a concentration of 1 mg/mL.

Only traces of cadaverine (at LOQ level) and tyramine (at LOD level) can be observed in the real sample at t_0_.

Generally, meat and meat-based products are usually subjected to BAs production due to their high content of amino acids and proteins; thus, proteolytic activity can arise during processing, storage, and aging, depending on changes to pH, sodium chloride presence, dehydration, and microbial activity [6]. For these reasons, BAs are widely studied and monitored as freshness markers or indicators of quality deterioration [6,20]. 

Table 3 reports the inferred concentrations in a beef sample at t_0_, limits of detection and quantification, recovery, and accuracy of measurements obtained for the same sample spiked with mix standards at different concentration levels.

Samples of shredded beef, wrapped in commercial PE film for food preservation (control, CTR), in SA and SA/GPPE film, each stored in a domestic refrigerator at 8 °C for different intervals of time (2 and 6 days), as reported in the Section 3, were SPME/GC–MS analyzed following the developed protocol. Figure 3 shows the concentration levels of the six selected BAs for each meat sample, wrapped in the different packaging materials corrected for the weight loss (about 5% for all samples, as expected by literature [21], at different storage times.

As evident from observing Figure 3, the BA levels in each beef sample wrapped in the different packaging materials increased during the time, especially for commercial PE film preserved samples (CTR). SA and SA/GPPE coatings, on the contrary, seemed to better preserve meat. No statistical significative difference (*t*-test, *p* < 0.05) between BA levels quantified in meat samples preserved in SA and SA/GPPE films was always registered. Moreover, the BA increase during the storage time seemed to be correlated with the pH rising of homogenized meat. After 6 days of storage in PE commercial film, it registered the highest pH value (7.30 ± 0.08), while SA and SA/GPPE showed lower values (5.38 ± 0.07) near the pH value measured for fresh meat (5.53 ± 0.04).

### 2.2. DSC Analysis: Assessment of the Beef Meat Samples Aging

To obtain more information, DSC analysis was then performed.

It is important to highlight that the beef muscle contains three types of water that differ from each other according to the freedom degree: bound water when it is involved in the interactions with hydrophilic groups of proteins, immobilized water when it is strongly held in the muscle structure, and free water when it is weakly linked to meat mainly by surface forces and can be easily released [22].

The aging of meat usually produces an increase in the amount of immobilized water due to the degradation of the myofibrillar, cytoskeletal, and intramuscular collagen proteins; the result is that the meat muscle structure is looser, rendering the capillary space more accessible to water [22].

Consequently, the amount of immobilized water can be indirectly used as a measure of the aging process and, therefore, of meat preservation. For this purpose, DSC analyses were performed. Figure 4 shows the thermograms of fresh meat samples and those preserved for 2 and 6 days when wrapped in CTR, SA, and SA/GPPE, respectively.

In detail, the DSC thermogram of the fresh meat sample showed the presence of two well-defined endothermic bands at around 80 and 100 °C associated with free and immobilized water, respectively. It was also possible to observe that the amount of immobilized water was greater than the free one. This effect could be related to an upcoming aging process in fresh meat, which could explain the presence of cadaverine and tyramine (Figure 2). The storage of the samples for 2 days in CTR, SA, and SA/GPPE produced important changes. The separation of the two endothermic peaks associated with free and immobilized water disappeared. An important increase in the peak associated with the immobilized water was observed in the case of the sample stored in CTR. The sample stored in SA also showed an increase in the peak associated with immobilized water, but not as intense as the one observed when studying the sample stored in PE film.

The presence of GPPE in the packaging material further reduced this effect, suggesting, for the two natural packaging materials, a retarded aging meat process as previously evidenced by SPME/GC–MS BA profile quantification (Figure 3).

The results associated with samples stored for 6 days confirmed the findings already observed after 2 days of storage time. The thermograms of all samples showed a further increase in the intensity of the peak associated with the immobilized water. This effect was less pronounced only in the sample stored in SA/GPPE, and there was still the presence of free water, as evidenced by the large band mainly located in the temperature range in which this type of water is expected.

### 2.3. Microbiological Evaluation of Beef Meat Samples

The effect of different films on the microbial growth in packaged fresh beef meat stored for 2 and 6 days at 8 °C was evaluated, and the results are reported in Figure 5.

In detail, TVC and psychrotrophic bacteria were studied to predict meat spoilage caused by biological agents. The initial level of microorganisms in the fresh beef sample was approximately 5.13 ± 0.25 and 5.00 ± 0.26 Log CFU/g for TVC and psychrotrophic bacteria, respectively. Cell density indicated the good quality of the used beef meat [23].

During storage, both TVC and psychrotrophic bacteria in the beef samples packaged with different films increased gradually (Figure 5) but with different rates.

Although after 2 days of storage, the cell density of TVC (5.87 ± 0.29, 5.86 ± 0.30, and 5.77 ± 0.28 Log CFU/g, for PE, SA, and SA/GPPE, respectively) and psychrotrophic (6.25 ± 0.31, 6.27 ± 0.32, and 6.66 ± 0.33 Log CFU/g, for PE, SA, and SA/GPPE, respectively) bacteria was equal for three meat packaged samples, after 6 days of storage at 8 °C an important difference was observed.

The TVC values of the SA and SA/GPPE samples were still acceptable (6.34 ± 0.32 and 6.30 ± 0.31 Log CFU/g, respectively). The same acceptable results were also collected for psychrotrophic bacteria.

After 6 days of cold storage, cell density detected was about 6.47 ± 0.32 and 6.77 ± 0.34 for SA and SA/GPPE, respectively. Otherwise, the increase of TVC and psychrotrophic bacteria in the PE packaged sample was significantly higher. Indeed, the cell density collected was about 7.03 ± 0.35 and 8.59 ± 0.43 Log CFU/g for TVC and psychrotrophic bacteria, respectively.

It has been reported that the microbiological limit for TVC in meat is about 6.7 log CFU/g [24]. Usually, the beginning changes in organoleptic properties (smell, taste, and appearance) are associated with high cell density (10^7^–10^8^ CFU/g) [25]. These results demonstrated that both the proposed lab-made films, SA and SA/GPPE, used as a coating material on beef meat samples during cold storage, had delayed the growth of TVC and, above all, of psychrotrophic bacteria.

As is well known, psychrotrophic or psychrophilic organisms such as Pseudomonas can grow on fresh meat [26]. In detail, psychrotrophic Pseudomonas spp. are the main microorganisms that cause spoilage of meat stored in refrigerated conditions. The increase of the psychrotrophic bacteria counts on the meat packaged and in cold storage conditions could depend on the presence of oxygen. On the contrary, the absence of oxygen, thanks to the presence of coating, can contribute to preventing the growth of Pseudomonas spp., being aerobic obligate [27,28].

At the end of storage incubation (9 days), the CTR sample still had the highest density (9.39 ± 0.46 and 10.63 ± 0.53 Log CFU/g for TVC and psychrotrophic bacteria, respectively) compared to SA (8.66 ± 0.43 and 8.21 ± 0.41 Log CFU/g for TVC and psychrotrophic bacteria, respectively) and to SA/GPPE (8.51 ± 0.42 and 8.02 ± 0.40 Log CFU/g for TVC and psychrotrophic bacteria, respectively) samples. These results highlighted a good/interesting capacity to reduce the rate of bacterial growth in beef meat over time.

On the basis of experimental evidence hereafter reported, the ability to inhibit at least up to 6 days at 8 °C the meat anaerobic degrative processes exhibited by SA and SA/GPPE coatings can be undoubtedly appreciated with respect to commercial PE packaging film.

## 3. Materials and Methods

### 3.1. Active Packaging Coatings

#### 3.1.1. Chemicals

Alginic acid sodium salt from brown algae (medium viscosity); CaCl2 anhydrous, granular, ≤7.0 mm, ≥93.0%; glycerol (anhydrous, reagent grade, having ≥99.5% purity), were purchased from Sigma-Aldrich (St. Louis, MO, USA).

Grape pomace waste was received from a local supplier, “L’Archetipo, Contrada Tafuri sp. 21, km 7, Castellaneta, Taranto (Italy)”, and stored at −19 °C before use.

Distilled water obtained by the Millipore Milli-Q Integral 5 Water Purification System was used to prepare the SA hydrogel and obtain the polyphenolic extract.

#### 3.1.2. GPPE Extraction

GPPE was prepared by adding 50 g of the as-received mixed grape waste (seeds, skin, and stems) into 1500 mL of distilled water, previously boiled for 30 min. Subsequently, to remove the coarse solid residual, vacuum filtration was accomplished. Then, the derived aqueous extract was centrifuged with a Thermo Fischer Scientific (Waltham, MA, USA) Heraeus Multifuge X3R centrifuge and stored at −19 °C before its use. [13,14]

#### 3.1.3. SA and SA/GPPE Film Preparation

The SA hydrogels (1% *w*/*v*) were prepared by solubilizing the alginic acid sodium salt powder in distilled water. CaCl_2_ (2.5% *w*/*v*) and glycerol (1 mL/100 mL), as a plasticizer, were added. The mixture was stirred for 24 h at room temperature to ensure the complete alginate dissolution and subsequently transferred to round petri plates for letting the solvent dry in an oven at 60 °C for 24 h. Free-standing, water-soluble solid films were obtained that, after soaking with a CaCl_2_ 5% (*w*/*v*) solution for 10 min, became water resistant. Thus, the realized films appeared not water-soluble, as carefully described in our recent work [13].

The same procedure was applied for preparing composite films containing the GPPE, added at a concentration of 40% (*v*/*v*) during the hydrogel preparation. Before the use, the obtained films were stored at −19 °C [13,14].

### 3.2. BAs Determination

#### 3.2.1. Chemicals

Six BAs standards (butylamine, cadaverine hydrochloride, isobutyl amine, isopentylamine, putrescine dihydrochloride, tyramine hydrochloride) and one internal standard (1–7 diaminoheptane, IS) were used. All reagents were purchased from Sigma-Aldrich and were >99% pure, except for cadaverine hydrochloride and the IS (98%). IBCF (Sigma-Aldrich) was used as a derivatizing agent.

Stock solutions of BAs (1 mg/mL) were prepared in sterile filtered ultrapure water (SFUW, Sigma-Aldrich) and stored in a refrigerator (8 °C). Working solutions of analytes alone or in a mixture were made daily by diluting stock solutions with SFUW.

IS solution was obtained at concentrations of 0.2 mg/mL and stored in the refrigerator for the time of trials.

#### 3.2.2. SPME/GC–MS Protocol

The SPME device, consisting of a manual holder and a polyacrylate (PA; film thickness diameter 85 μm), was obtained from Sigma-Aldrich. The fiber was conditioned in the GC injector as suggested by the supplier before analysis. Then, it was exposed for 40 min into a 1.7 mL clear glass vial sealed with a PTFE (polytetrafluoroethylene) septum containing a Teflon-coated magnetic stir bar (4 × 10 mm) (Sigma-Aldrich) and 1.5 mL of a sample solution obtained in accordance with Aresta et al. [19]. Briefly, 0.05 mL of standard solutions or supernatant homogenized meat sample was transferred into the sealed vial containing 1.43 mL of a sterile 15% NaCl solution. IS (7.5 μL) was added employing a syringe, and the pH of the resulting solution, in accordance with [29], was adjusted to 12 with 4 N NaOH (6.0 μL). Finally, 7.5 μL of the derivatizing agent IBCF was added. The vial was immediately manually shaken for 2 min at room temperature, and the SPME fiber was exposed to the solution for 40 min. Finally, fiber was carefully removed from the vial and directly inserted into the GC injector for 10 min for the GC–MS analysis.

The GC–MS system was a Finnigan TRACE GC ultra-gas chromatograph (Thermo Fisher Scientific, Waltham, MA, USA) equipped with a split/splitless injector, interfaced to an ion trap MS (Finnigan Polaris Q, Thermo Fisher Scientific). The capillary column was a Sigma-Aldrich SPB-5 fused silica (30 m, 0.25 μm i.d., 0.25 μm film thickness) with helium (purity > 98%, Rivoira, Bari, Italy) as carrier gas (flow rate 1 mL/min). The temperature of the transfer line was 300 °C, while the injector (splitless mode for 1 min) was kept at 250 °C. The oven temperature program was: 100 °C held for 1.20 min; ramp 1:10 °C/min from 100 °C to 150 °C; 160 °C; ramp 2:23 °C/min from 280 °C held for 12 min; ramp 3:25 °C/min to 300 °C; held for 10 min. The mass spectrometer was operated in the electron impact positive ion mode (EI+) with the ion source temperature set at 250 °C. The electron energy was 70 eV, and the filament current was 150 μA. Detection of analyte targets was made from extracted ion chromatograms (XICs) obtained in total ion current mode (TIC, m/z range 50–450). The extracted ions were 84, 107, 112, 118, 120, 130, 132, and 170 m/z based on the retention times (RT) and characteristic m/z ions for selected BAs and IS, as shown in Table 1.

To remove carry-over, fiber was always subjected to a second thermal desorption after each chromatographic run. All experiments were performed in triplicate.

To validate the described method, calibration curves were obtained analyzing BA working solutions in the concentration range 0.06–300 μg/mL, corresponding to 0.002–10 μg/mL in the vial. The limits of detection (LOD) and limits of quantification (LOQ) were determined by LOD ≅ (3·sda)/b and LOQ ≅ (10·sda)/b, where sda is the standard deviation of the y-intercept and b is the slope of the regression line. The within-day (*n* = 3) and between-days (*n* = 3 over 5 days) percentage relative standard deviations (RSD %) were calculated at three levels (0.05, 0.25, and 2.5 μg/mL in the vial), analyzing solutions daily prepared by the same working solutions stored at 8 °C.

### 3.3. Real Samples

Minced fresh beef (about 35–50 g), purchased from a local supplier (Bari, Italy), was cut by a sterile knife to reach approximately the same weight (about 5 g) and then wrapped in industrial PE film, in lone SA film, and in SA/GPPE film. Specifically, two round films were superimposed and used for packing the meat samples, placed between them, as depicted in Figure 1. All samples were kept in a domestic refrigerator at 8 °C and stored for different time intervals (0, 2, and 6 days). Each sample was processed for the determination of BAs by the proposed SPME/GC–MS protocol for pH measurements, differential scanning calorimetry (DSC) analysis, and biological tests.

-SPME/GC–MS sample: 0.25 g of beef meat was weighed directly in 15 mL vials (Sigma-Aldrich) and spiked with 0.15 mL of IS solution (0.2 mg/mL). Then, 5 mL of 2,4,6-trichloroanisole (TCA) 0.3 N was added in each vial, and a homogenized was readily obtained using Homogenizer Lab Gen 7 (Cole-Parmer Instrument Company, Vernon Hills, IL, USA) operated to a medium speed for 5 min in ice. The homogenized was centrifuged at 2.370 g for 5 min at 22 °C (Hermele Z216MK, Labor Technik, Wasserburg, Bodensee, Germany). The supernatant was transferred in a hermetically sealed sterile vial. Afterward, 0.05 mL was transferred into the sealed vial for the quantitative evolution study, following the SPME/GC–MS protocol (Section 3.2.2).

To estimate sensitivity, recovery, and accuracy of measurements in beef, mix standard solutions containing the six selected BAs at twice and five times the estimated concentrations, or the LOD levels, were added to each sample, and then the ratio between the peak area in the spiked sample and the standard solutions at the same concentrations was calculated.

MedCalc Software Ltd. 22.014 version was used for performing statistical analysis (F-test).

-pH measurements: 0.25 g of beef meat were homogenized in 10 mL of distilled water under constant stirring. Measurements were performed at each storage time (0, 2, and 6 days) for all samples wrapped in the three different types of selected protective films using a calibrated pH meter (Hanna Instruments, Padova, Italy).-DSC analysis: experiments were performed with a Q200 instrument (TA Instruments, New Castle, DE, USA) in the range from 25 °C to 300 °C, according to experimental conditions described in [30].

### 3.4. Microbiological Analysis

The microbiological analysis was evaluated on fresh beef meat samples wrapped in the different coatings (CTR, SA, and SA/GPPE) after 2 and 6 days of storage at 8 °C. 1 g of each beef sample was weighed and transferred aseptically into falcon containing 9 mL of sterile saline solution (0.9% NaCl bacteriological grade; Oxoid, Basingstoke, UK). The mixture was homogenized using Homogenizer Lab Gen 7 for 2 min at room temperature. The microbial enumeration was made by serial decimal dilutions of the samples, prepared in 0.9% NaCl, and poured onto the corresponding Petri dishes. Total viable count (TVC) and psychrotrophic bacteria were enumerated by using Plate Count Agar (PCA; Oxoid, Basingstoke, UK), incubated at 30 °C for 48 h and 4 °C for 7 days, respectively. Microbiological counts were expressed as logarithms of the number of colony-forming units per g sample (Log CFU/g) [31]. Microbiological analyses were carried out in duplicate.

### 3.5. Statistical Analysis

All the viability experiments were achieved in triplicate for each treatment. Results were presented as mean values with standard deviation (SD). Statistical analysis was performed by analysis of variance to one-way (ANOVA) with Statistica for Windows (Statistica 10, Windows) using Tukey’s test, and significant difference was considered at *p* < 0.05.

## 4. Conclusions

In this work, the optimization of an analytical protocol to study the preserving activity of an alternative, biodegradable, sodium alginate-based packaging material enriched with GPPE useful for high protein content food (e.g., red meat) storage was proposed.

The production over time of six biogenic amines (butylamine, cadaverine, isobutylamine, isopentylamine, putrescine, and tyramine), “sentinels” of the meat putrefactive process, was followed during the storage time in a domestic refrigerator up to 6 days, by SPME/GC–MS. The obtained results showed that the production of BAs was reduced in the case of beef meat samples stored in SA and SA/GPPE films if compared to those stored in a commercial polyethylene film (CTR).

The changes in pH and weight values and immobilized/free water content, for the same time intervals, as well as the bacterial growth, were also evaluated. For comparison, the evolution of beef samples wrapped in commercial PE industrial film and lone sodium alginate coating was followed.

The optimized analytical protocol results are useful for demonstrating that the SA and SA/GPPE film exhibited better efficiency in retarding anaerobic meat degradative processes at 8 °C, at least up to 6 days, if compared to the commonly used commercial PE-based polymeric ones.

Additionally, our previous work suggested that grape pomace extract is rich in polyphenols having high antioxidant properties [13]. Therefore, future studies will be carried out to assess the possible antioxidant performance of the developed SA/GPPE packaging material when storing protein and lipid-rich foods, vegetables, and fruits.

## Data Availability

All of the data is contained within the article.

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
