# Peer review of "Effect of a Composite Alginate/Grape Pomace Extract Packaging Material for Improving Meat Storage"

_ijms, 2023, doi:10.3390/ijms242115958_

Round 1
Reviewer 1 Report
Comments and Suggestions for Authors
Thr authors have presented an interesting study on the use of a film of alginate and grape pomace for preservation of meat in a refrigerator. The authors have presented a further step (focusing on the application of the material for the food inductry) of their research on this film which was previously published. As explained by the author, the primary goal is to make a material alternative to non-biodegrdable plastics.
I think the paper is interesting and minor improvements might make the paper better.
1. Could the author mention whether they have carried out short and long term stability studies of the film? Furthermore, as meat is often stored in a freezer, it would be interesting to know how the material behave at -20C. This could be included in future work.
2. Please Rephrase the first sentence of paragraph 3.1.2 as it isn't clear.
3. Could the authors explain how it is possible that the material is water resistant? I have read the previous paper, and you could re-explain your theory here. Furthermore, I don't believe the material is water resistant, but it is stable in water under certain conditions. I am quite confident in a mid-term it will degrade in water.
For Fig. 3 and 5 please report the stats on the graphs.
Comments on the Quality of English LanguageMinor English editing
Author Response
#Reviewer 1
Thr authors have presented an interesting study on the use of a film of alginate and grape pomace for preservation of meat in a refrigerator. The authors have presented a further step (focusing on the application of the material for the food inductry) of their research on this film which was previously published. As explained by the author, the primary goal is to make a material alternative to non-biodegrdable plastics.
I think the paper is interesting and minor improvements might make the paper better.
Comment:
- Could the author mention whether they have carried out short and long term stability studies of the film? Furthermore, as meat is often stored in a freezer, it would be interesting to know how the material behave at -20C. This could be included in future work.
Reply:
We would like to acknowledge the reviewer for His/Her positive feedback, and the suggestions for improving our manuscript.
In our previous work (reference 13. Gubitosa, J.; Rizzi, V.; Marasciulo, C.; Maggi, F.; Caprioli, G.; Mustafa, A.M.; Fini, P.; De Vietro, N.; Aresta, A.M.; Cosma, P. Realizing eco-friendly water-resistant sodium-alginate-based films blended with a polyphenolic aqueous extract from grape pomace waste for potential food packaging applications. Int. J. Mol. Sci. 2023, 24, 11462-11480. https://doi.org/10.3390/ijms241411462) we have already described and discussed the films stability by performing swelling tests in water-based solutions, also when varying the pH, temperature, ionic strength, and light sources, although without further specifying the period during which their stability was monitored. Anyway, the films appeared stable for longer time (3-4 months) at different temperatures, not pH-responsive, and photostable after irradiation with Neon-, UV-, and solar simulator lamps. Only highly concentrated salt-based solutions negatively affected the films stability, determining a large swelling. However, it should be highlighted that long storage time conditions could be considered not so realistic and suitable for beef meat samples.
Work is in progress in our laboratory for testing the realized composite films behavior when stored at -20°C to pack meat samples for longer time.
Comment:
- Please Rephrase the first sentence of paragraph 3.1.2 as it isn't clear.
Reply:
The sentence has been revised, accordingly.
More specifically, it has been reported in the manuscript as: “GPPE was prepared by adding 50 g of the as-received mixed grape waste (seeds, skin, and stems) into 1500 mL of distilled water, previously boiled for 30 min.”
Comment:
- Could the authors explain how it is possible that the material is water resistant? I have read the previous paper, and you could re-explain your theory here. Furthermore, I don't believe the material is water resistant, but it is stable in water under certain conditions. I am quite confident in a mid-term it will degrade in water.
Reply:
As already reported above in our previous reply to your comment, a modified protocol for preparing water-resistant SA films by adopting both external and internal gelation methods have been extensively discussed in our previous paper; for this reason, we avoided to report the theory also in the present work. Anyway, according to the reviewer’s suggestion, the manuscript has been revised, and some more information about this topic has been added for briefly explain the approach followed for realizing water-resistant films stable for longer time, even when exposed to different conditions. Our theory has been revised and reported in the manuscript as: “A modified protocol was followed for preparing water-resistant SA-based films by adopting both external and internal gelation methods. More specifically, as first step, the SA hydrogel was mixed with a 2.5% (w/v) CaCl2 solution, and then introduced in an oven for removing water; thus, the solid-state film formation occurred. The obtained free-standing SA-based films were placed in contact with a 5% (w/v) CaCl2 solution.
Depending on the applications, both methods would offer SA-based films constituted by polymer chains differently packed and water resistant, if compared with those realized without using CaCl2. [13] In this particular case of study, two phenomena occurred simultaneously: (i) hydration, and (ii) the further crosslinking reaction with Ca2+, favored by the high CaCl2 concentration. [13] Strong coulombian interactions occurred between Ca2+ ions and the alginate carboxylic groups compacting the polymeric network and making it less hydrophilic and less permeable to water. [13]”
Comment:
For Fig. 3 and 5 please report the stats on the graphs.
Reply:
Figures 3 and 5 have been revised, accordingly.
In detail, Figure 3 has been modified reporting, as required, the stats on graph obtained by employing the MedCalc Software Ltd. 22.014 version (F-test). In the “Materials and method” section, the information about the used software has been added.
Figure 3. BAs concentration levels, at different storage times at 8 °C, for beef samples packed in CTR, SA, and SA/GPPE films, respectively. (*) marks samples characterized by significantly different (F-test, p < 0.05) BAs levels.
Figure 5 and the related caption have been modified by indicating the significantly different viable counts for the investigated samples at different storage time as requested by the Reviewer.
Figure 5: Cell density (Log CFU/g) of total viable count (TVC) (orange bar) and psychotrophic (blue bar) bacteria of fresh and beef meat samples packaged with CTR, SA and SA/GPPE films, after 2 and 6 days of storage at 8 °C. For each storage time, the different letters (a, b, c) indicate the significant differences (P ≤ 0.05).
A paragraph has been added in the “Materials and Methods” section of the revised manuscript:
3.5 Statistical analysis
All the viability experiments were achieved in triplicate for each treatment. Results were presented as mean values with standard deviation (SD). Statistical analysis was performed by analysis of variance to one way (ANOVA) with Statistica for Windows (Statistica 10, Windows) using Tukey's test and significant difference was considered at P<0.05.

Reviewer 2 Report
Comments and Suggestions for Authors
Dear Authors,
My comments bellow.
Lines 34-35: Please, reformulate this sentence because it is not clear.
Do you have any pictures ilustrated the experiment? Packaging?
Lines 264-274: How and where did you store the films? Conditions? How did you applicate those films and onto what? Or did you wrapped?
Please add more information or a section about the packagin preparation.
Results and discussion: rebuild this section please, more conclussions are needed.
English should be improved.
Comments on the Quality of English Language
English should be improved.
Author Response
#Reviewer 2
Dear Authors,
My comments bellow.
Comment:
Lines 34-35: Please, reformulate this sentence because it is not clear.
Reply:
The sentence has been revised, accordingly. More specifically, it has been reported in the manuscript as: “Amines are low molecular weight organic bases produced in animals, plants, or microorganisms after specific metabolic pathways”.
Comment:
Do you have any pictures ilustrated the experiment? Packaging?
Reply:
According to the reviewer’s suggestion, a picture displaying the experimental steps followed for preparing the films and packing the beef meat samples has been added in the revised paper and reported in Scheme 1.
Comment:
Lines 264-274: How and where did you store the films? Conditions? How did you applicate those films and onto what? Or did you wrapped?
Reply:
The lines 264-274 indicated by the reviewer in His/Her comment are referred to the SA and SA + GPPE films preparation protocol. The information about the required experimental conditions explanations have been reported in the “Materials and methods” section, sub-section “3.3 Real samples”, as follows: “Minced fresh beef (about 35-50 g), purchased from a local supplier (Bari, Italy), was cut by a sterile knife for approximately reaching the same weight (about 5 g), and then wrapped in industrial PE film, in lone SA film, and in SA/GPPE film. All samples were kept in a domestic refrigerator at 8 °C and stored at different time intervals (0, 2, and 6 days).”
Anyway, before packing the meat samples, the films were stored at -19°C.
Comment:
Please add more information or a section about the packagin preparation.
Reply:
The protocol followed for preparing the studied SA-based packaging material has been already reported in the “Materials and methods” section, sub-section “3.1.3. SA and SA/GPPE film preparation”, as follows: “The SA hydrogels (1% w/v) were prepared by solubilizing the alginic acid sodium salt powder in distilled water. CaCl2 (2.5% w/v) and glycerol (1 mL/100 mL) as a plasticizer, were added. The mixture was stirred for 24 h at room temperature to ensure the complete alginate dissolution, and subsequently transferred to petri plates for letting the solvent dry in an oven at 60 °C for 24 h. Free-standing, water-soluble solid films were obtained that, after soaking with a CaCl2 5% (w/v) solution for 10 min, became water resistant. The same procedure was applied for preparing composite films containing the GPPE, added at a concentration of 40 % (v/v) during the hydrogel preparation.”
Comment:
Results and discussion: rebuild this section please, more conclussions are needed.
Reply:
The “Results and discussion” section, as well as the “Conclusions” have been revised according to the reviewer’s suggestion.
Comment:
English should be improved.
Reply:
The English form has been revised, accordingly, for improving the quality of the manuscript.

Reviewer 3 Report
Comments and Suggestions for Authors
1) The title is very long and unintelligible, I recommend to shorten it and more specify. The dot should not be at the end.
2) Is the main topic - alternative food packaging, active packaging, BA´s determination, food spoilage, foos shelf-life? The authors should be more precise in their point of view of this paper. It seems to be inconsistent, needs to be improved.
3) Results - the Figure 5 - it is not cell density, it can be named like viable counts, or only Log CFU/g.
4) Did you measure also total viable counts of anaerobes? Probably it will be interesting, especially in discussion with the content of biogenic amines.
5) the reference 17 is redundant
6) the tyramine content can be more discussed in the discussion part
7) the aim of the work should be more clear
Author Response
#Reviewer 3
Comment:
1) The title is very long and unintelligible, I recommend to shorten it and more specify. The dot should not be at the end.
Reply:
The title has been revised according to the reviewer’s suggestion; a new title has been assigned to the revised version of the manuscript, without reporting the dot at its end, as follows: “Effect of a composite alginate/grape pomace extract packaging material for improving meat storage”.
Comment:
2) Is the main topic - alternative food packaging, active packaging, BA´s determination, food spoilage, foos shelf-life? The authors should be more precise in their point of view of this paper. It seems to be inconsistent, needs to be improved.
Reply:
According to the reviewer’s suggestion, the main topic of the work has been better highlighted.
Comment:
3) Results - the Figure 5 - it is not cell density, it can be named like viable counts, or only Log CFU/g.
Reply:
We thank the Reviewer for the suggestion. The y axis title has been modified in viable counts (Log CFU/g).
Comment:
4) Did you measure also total viable counts of anaerobes? Probably it will be interesting, especially in discussion with the content of biogenic amines.
Reply:
We thank the Reviewer for the suggestion. We agree about the fact that the contribution of anaerobic microorganisms may be interesting for making correlations with the production of biogenic amines. Indeed, we are currently working on cultivating anaerobic microorganisms and we will report the results in a future paper, along with further microbiological assessments on other foodstuff, not only on beef meat samples.
Comment:
5) the reference 17 is redundant
Reply:
After carefully checking the manuscript, we have noted that the reference 17 has been reported for one time, and only in the introduction. Anyway, the reference 17 has been replaced by 13 and 14 ones.
Comment:
6) the tyramine content can be more discussed in the discussion part
Reply:
Generally, meat and meat-based products are usually subjected to BAs production due to their high content in amino acids and proteins; thus, proteolytic activity can arise during processing, storage, and ageing, depending on changing of pH, sodium chloride presence, dehydration, and microbial activity. [6] For these reasons, BAs are widely studied and monitored as freshness markers or indicators of quality deterioration. [6, 20] Tyramine is one of those, but not the lone.
Comment:
7) the aim of the work should be more clear
Reply:
As suggested by the reviewer, in the revised version of the manuscript the aim of the work has been further clarified.

Round 2
Reviewer 3 Report
Comments and Suggestions for Authors
Thank you for excepting the recommendations.